# Complexes of Glucarolactones with Water-Soluble Copolymers of N-Vinylpyrrolidone with N-Vinylamine as Inhibitors of β-Glucuronidase Efficacy

**DOI:** 10.3390/polym14010105

**Published:** 2021-12-28

**Authors:** Valerii D. Krasikov, Yulia G. Santuryan, Irina I. Malahova, Alexey G. Ivanov, Nikolay I. Gorshkov, Evgenii F. Panarin

**Affiliations:** Federal State Budgetary Institution of Science Institute of Macromolecular Compounds, Russian Academy of Sciences (IMC RAS), 199004 Saint Petersburg, Russia; jsant@hq.macro.ru (Y.G.S.); iimalahova@mail.ru (I.I.M.); alexey.ivanov@bk.ru (A.G.I.); ngorshkov@mail.ru (N.I.G.); panarin@hq.macro.ru (E.F.P.)

**Keywords:** water-soluble copolymers, N-vinylpyrrolidone/N-vinylamine, lactones of D-glucuronic acid, bladder tumor, polymer complexes of glucarolactones, inhibition of β-glucuronidase

## Abstract

Water-soluble complexes of N-vinylpyrrolidone/N-vinylamine copolymers with lactones of D-glucuronic acid were obtained and characterized by chromatographic, spectral, and hydrodynamic methods. The complexes efficiently inhibited the enzyme β-glucuronidase that causes the appearance of bladder tumors. The products demonstrated prolonged action and were stable during storage.

## 1. Introduction

An important modern direction in the development of antitumor preparations is enhancing the therapeutic index of drugs. Cytotoxic substances are accumulated not only in tumor tissues, but also in non-target organs, leading to the intoxication of an organism. The therapeutic index of a preparation may be improved by binding the drug to a macromolecule, which leads to an increase in the local concentration of this therapeutic agent in target pathological cells. As a result, toxicity and dose of a preparation can be sharply reduced. In recent decades, the attention of researchers has been focused on the use of synthetic polyfunctional water-soluble and biocompatible polymers as carriers for biologically active compounds [1,2,3,4].

The tasks of modern pharmaceutics include obtaining antitumor drugs with prolonged action. The principal possibility of the creation of prolonged-action drugs is based on the modification of target anionic drugs with synthetic cations [5,6]. Polycations possess the capability of selective binding to the surface of tumor cells (which carry negative charges); thus, it is possible to use polycations in “targeted therapy” for diagnostics and treatment of oncological diseases [7,8]. Polycations increase the permeability of cell membranes, act as interferon inductors, form complexes with DNA, and demonstrate antimicrobial activity [9].

Water-soluble carbochain polymers based on poly-N-vinylamides are frequently used as such polycations in various areas of pharmaceutics and medicine [1,9,10,11,12].

Cyclic N-vinylamides (N-vinylpyrrolidone, N-vinylcaprolactam) and open-chain Nvinylamides (N-vinylformamide, N-vinylacetamide) are sufficiently well studied [13,14]. Homopolymers and copolymers of N-vinylamides containing monomers with protected amino groups currently attract broad attention; the copolymers (e.g., water-soluble copolymers of N-vinylpyrrolidone (N-VP) with N-vinylamine (N-VA)) can be employed in the synthesis of cationic carriers for biologically active compounds (BAC) [9,15,16]. At the same time, it was found that polymers with amino groups possess immunostimulating properties that may cause a strong immune response towards an introduced antigen, depending on the content of ionogenic groups [17,18]. This fact should be taken into account when creating prolonged-action drugs.

It is known [19] that bladder tumors are caused, among other things, by the accumulation of carcinogenic metabolites in the bladder mucous membrane. It has been established that carcinogenic substances (both exogenous and endogenous) are excreted through kidneys in the form of water-soluble conjugates with D-glucuronic acid. However, their excretion path (bladder) contains the enzyme β-glucuronidase (β-GA) [20], which splits these conjugates and releases D-glucuronic acid and a free lipid-soluble carcinogenic metabolite [21]. The metabolite becomes accumulated in the bladder mucous membrane and causes the consequent development of papillomas and malignant growths [21,22]. The activity of β-GA in various tumor cells is significantly (several times) higher than its activity in normal cells [23].

Currently, selective inhibitors of β-glucuronidase are known [22,23,24,25]. A number of research groups have demonstrated that some derivatives of D(+)-glucaric acid [26,27,28,29,30], particularly 1,4- and 6,3-glucarolactones (1,4-GL and 6,3-GL) [31,32], serve as specific competitive inhibitors of β-GA. These inhibitors suppress activity of β-GA and prevent decomposition of the carcinogen–glucuronic complex. Thus, excretion of carcinogenic metabolites in the water-soluble form is provided; this process prevents the accumulation of carcinogens in the organism, malignization of bladder epithelium, and development of epithelial tumors (papillomas and carcinomas) [26,33].

Experimental and clinical studies showed that after peroral administration of glucarolactones to patients, only short-term (4–5 h) inhibition of β-GA contained in urine was provided, since lactones are rapidly cleared from the organism [34]. To reach the long-lasting inhibitory effect of lactones, it is necessary to develop their drug formulations with prolonged action. One approach to the creation of prolonged-action drugs is modifying these substances with polymers [5,35].

Graft copolymers of several glycomonomers (including glucarolactones) with acrylamide and styrene have been synthesized; the inhibiting activity of the conjugates has been studied, and mechanisms of transformations of glycoside rings grafted onto the polymer chain and their dependence on pH have been investigated [36,37,38].

Physico-chemical properties of the preparations based on water-soluble polymers that inhibit β-glucuronidase have been studied insufficiently. Thus, the goals of the present study included: (i) the synthesis of water-soluble complexes of glucarolactones (inhibitors of enzyme β-glucuronidase) with [N-VP]_n_-[N-VA]_m_ copolymer carriers; (ii) studies of the structure and physico-chemical properties of the complexes by chromatographic (thin-layer, size exclusion chromatography) and spectral (NMR, FT-IR, UV spectroscopy) methods, by molecular optics and hydrodynamics (velocity sedimentation, molecular diffusion, flow birefringence, dynamic light scattering (DLS)); (iii) estimation of their specific biological activity.

## 2. Materials and Methods

### 2.1. Materials

Reagent grade and special purity grade substances (Sigma-Aldrich, St Louis, MO, USA), NevaReaktiv, (St Petersburg, Russia) were used in the experiments. Aqueous solutions were prepared in deionized water with resistivity equal to 18 MΩ/cm purified using the Simplicity Millipore (Millipore, Molsheim, France).

Monomers of N-vinylpyrrolidone and N-vinylformamide (Fluka, Germany) were purified by double distillation under vacuum over granulated KOH. The fractions with b.p. 342 K/3399 Pa, n_D_^20^ = 1.5117 and 335 K/3399 Pa, n_D_^20^ = 1.4960, respectively, were taken.

### 2.2. Instruments and Measurements

Electronic spectra of samples were registered in the wavelength range from 190 to 700 nm with a UV-1280 spectrophotometer (Shanghai SEN Technology Co., Ltd., Shanghai, China); the samples were dissolved in bi-distilled water and put in quartz cuvettes 1 cm thick.

IR spectra were measured in KBr pellets using an IRAffinity-lS FTIR spectrometer (Shimadzu, Kyoto, Japan) in the range 400–4000 cm^−1^.

The ^1^H NMR spectra were taken in deuterated water using BrukerAvance II-500 WB spectrometer (Geretsried, Germany).

Thin layer chromatography (TLC) was performed using PTSH-P-A analytical planar plates (JSC “Imid”, Krasnodar, Russia). In all cases, chromatograms were visualized in iodine vapors. Concentrations of solutions for TLC varied from 2.5 to 5.0 mg/mL. Quantitative analysis of the TLC results was carried out with the aid of a “DenScan” video densitometer and the specialized software for image processing “Dens” (“Lenchrom” Scientific Center, Saint Petersburg, Russia) [39].

High-performance liquid chromatography (HPLC) was carried out using a Smartline liquid chromatograph (Knauer, Geretsried, Germany) equipped with differential refractometric detector and photometric detector with diode array. An Ultrahydrogel linear SEC column (7.8 × 300 mm) with a pre-column (0.6 × 40 mm, Waters, Milford, MA, USA) was used for analysis of the copolymers. The 0.2 M solution of sodium chloride in bi-distilled water served as an eluent; the temperature was 293 K; detection was performed at 200 nm and 230 nm.

The HPLC column was calibrated using the water-soluble poly (N-vinyl formamide) (PVFA) standards isolated earlier [40]. The sample concentration for HPLC was 1.5 mg/mL. Before experiments, the samples were filtered through 15 µm microfilters (Waters, Milford, MA, USA). The eluent mixtures were filtered through SYNPOR membrane filters (pore size 4.0 µm).

Intrinsic viscosity [*η*] was measured using an Ostwald capillary viscometer at 294 K. In the cases of weak saline solutions, the [*η*] values were determined from the initial slope of the ln(*ɳ*_r_) = f(c) dependence, where (*η*_r_) is the relative viscosity of a solution at concentration *c*.

Partial specific volumes $\bar{v}$ of copolymers were measured in a 2 mL glass pycnometer; the values were calculated by the formula Δ*ρ/*Δ*c = 1−ῡρ_o_* (where Δ*ρ* is the increment of solution density at concentration increment Δ*c*, and $\rho_{o}$ is the solution density).

Velocity sedimentation experiments were performed in a MOM 3180 analytical ultracentrifuge (Hungary) at the rotor speed of 4 × 10^4^ rpm (294 K). The sedimentation boundary was formed in a two-section cuvette by layering solvent over a solution. The cuvette thickness along the direction of light beam was 12 mm. Sedimentation coefficients *S* were determined within the concentration range *c* = (0.04–0.50) × 10^−2^ g/cm^3^.

Translational diffusion was studied with the use of the Tsvetkov polarization diffusometer [41,42]. Diffusion coefficients *D* were calculated on the basis of the slope of time dependence of dispersion of diffusion boundary 2σ^2^. These dependences were linear for all the studied samples. Since the measurements were carried out at very low concentrations, the diffusion coefficient *D* obtained at finite low concentration *c* = (0.04–0.06) × 10^−2^ g/cm^3^ was taken as the diffusion constant.

Hydrodynamic molecular masses were estimated from the data of sedimentation–diffusion analysis according to the Svedberg formula *M**_SD_*= (s_0_/*D*_0_)[R*T*/(l−$\bar{v}\rho_{o}$), where *R* is the universal gas constant, and *T* is the absolute temperature.

Average hydrodynamic radii of macromolecules (*R*_h_) were determined by dynamic light scattering (DLS) with the use of a Photocor Complex correlation spectrometer (light source: He–Ne laser (Coherent Inc., Santa Clara, CA, USA), power output: 20 mW, wavelength 632.8 nm) equipped with a programmed Photo Cor-FC correlator (ZAO Anteks, Russia, 288 channels). The correlation function was processed with the use of Dynals software (Helios, St.-Petersburg, Russia), which allows one to use the obtained diffusion coefficients *D* to calculate hydrodynamic radii of equivalent sphere *R*_h_ according to the Einstein–Stokes equation: *R*_h_ = k*T*/6*pD*h_0_, where *k* is the Boltzmann constant, h_0_ is the solvent viscosity, and *T* is the temperature. The scattering angle was 90° [43,44].

Weight average masses *(*$\bar{M}$*_w_*)* of a number of polymers were determined by static light scattering in 0.2 N aqueous solution of NaCl. The results were processed by the Debye method [45,46] using the data for the scattering angle of 90°, since asymmetry of light scattering in these solutions was not high.

### 2.3. Synthesis of [N-VP]_n_-[N-VFA]_m_ and [N-VP]_n_-[N-VA]_m_ Copolymers

Water-soluble copolymers of N-vinylpyrrolidone with N-vinylamine ([N-VP]_n_-[N-VA]_m_) were prepared by the two-stage method [47,48] involving removal of formyl protecting group by acidic hydrolysis [49]. First, the precursor-copolymer N-vinylpyrrolidone with N-vinylformamide ([N-VP]_n_–[N-VFA]_m_) was synthesized by free radical polymerization initiated by hydrogen peroxide [42,43]. Then, the protection was removed from [N-VP]_n_–[N-VFA]_m_ chains by treatment with orthophosphoric acid at 383 K for 24 h with the formation of vinylamine units [44].

After hydrolysis, formic acid was distilled off in the form of azeotropic mixture with water (composition of the azeotropic mixture: formic acid: 44.4 wt.%, water: 22.6 wt.%) at the boiling temperature (382 K). After distilling off formic acid, 45% aqueous solution of NaOH was added into the resulting concentrated solution of the [N-VP]_n_–[N-VA]_m_ copolymer that contained phosphoric acid; the mixture was stirred for 12 h. The solution of the copolymer in phosphate buffer was obtained.

Monitoring of residual monomer content during synthesis of the [N-VP]_n_–[N-VFA]_m_ copolymer, appearance of formic acid after removal of formyl protection from N-VFA units of the copolymer, and analysis of composition of polymer complex [N-VP]_n_–[N-VA]_m_ with 1,4- and 6,3-lactones of saccharic acid were performed by thin layer chromatography (TLC) [50].

Composition of the target copolymer (content of N-VA units) was determined photometrically from the intensity of absorption of its complex with 2,4,6-trinitrobenzene sulfonic acid (λ_n_ = 420 nm) [51,52].

Copolymer complexes [N-VP]_n_–[N-VA]_m_ were obtained in the molecular mass range of 27–54 kDa.

Characteristic spectral data are as follows.

^1^H NMR (500 MHz, D_2_O): δ, ppm, 39–1.83 (CH_2_CH (C_4_H_6_NO), br, 2H), 1.84–2.07 (NCH_2_CH_2_CH_2_CO, br, 2H), 2.08–2.50 (NCH_2_CH_2_CH_2_CO, br, 2H), 2.93–3.38 (NCH_2_CH_2_CH_2_CO, br, 2H), 3.41–3.86 (CH_2_CH (C_4_H_6_NO), br 1H).

^13^C NMR (500 MHz, D_2_O): δ, ppm, 17 (NCH_2_CH_2_CH_2_CO), 31 (NCH_2_CH_2_CH_2_CO), 35 (CH_2_CH (C_4_H_6_NO), 44 (NCH_2_CH_2_CH_2_CO), 52 (CH_2_CH (C_4_H_6_NO), 177 (C=O).

IR (KBr pellets) Lactam ring C=O 1642 cm^−1^.

In order to perform analytical control experiments, phosphates present in the [N-VP]_n_–[N-VFA]_m_ copolymers were transferred into OH^−^ form by preparative ion exchange chromatography using the EDE-10P anionite (Russia). Then, the product was dialyzed against water in the OrDial D14 dialysis membrane bag (pore size 1.0 kDa, Orang Scientific, Braine-l’Alleud, Belgium). The purified copolymers were isolated from water solutions by freeze-drying.

### 2.4. Synthesis of Lactones of Saccharic Acid

Individual lactones of saccharic acid were synthesized and isolated according to the following scheme: glucose was oxidized by 43% nitric acid, and acidic potassium salt of saccharic acid (C_6_H_9_O_8_K) was isolated [53]. Then, solution of acidic potassium salt was treated with cation exchanger KU-2 (Chelyabinsk Power Machine Building Plant, Chelyabinsk, Russia) in the H-form, and the solvent was partially removed using a Stegler RI-213 rotary evaporator). The residual D(+)-glucaric acid was removed by filtering off acidic potassium salt of the acid precipitated after alkalization of the concentrated solution with 45% solution of KOH.

The mixture of lactones enriched in 1,4-glucarolactone was obtained by extraction with aliphatic alcohols followed by evaporation of alcohols under vacuum at 298 K; 6,3- glucarolactone was isolated by crystallization from aqueous layer. The melting points of 1,4-glucarolactone (366 K) and 6,3-glucarolactone (422 K) corresponded to the reference data.

### 2.5. Synthesis of [N-VP]_n_-[N-VA]_m-_ β-GA Complexes

The polymeric complex between [N-VP]_n_-[N-VA]_m_ copolymers (400 mg) and the mixture of 1,4- and 6,3-glucarolactones (100 mg) was prepared by stirring their solutions in phosphate buffer (pH 4.75) (20 mL) for 40 h. The complexes were dialyzed against water in order to remove excesses of glucarolactones and then freeze or spray-dried. Formation and stability of the complexes were studied using chromatographic and spectral methods (analysis of free glucarolactones isolated after equilibrium dialysis).

### 2.6. In Vitro and Vivo Experiments

Activity of the obtained polymer complexes and degree of β-GA inhibition were estimated according to the Fishman (phenolphthalein) method [54]; sodium salt of phenolphthalein-β-D-glucuronidase (Koch-Light Lab, Haverhill, England, Great Britain) was used as a substrate. Concentration of free phenolphthalein was determined spectrophotometrically (pH = 10.2–10.3, wavelength λ = 540 nm). The enzyme activity was expressed in arbitrary units, i.e., the amount (mg) of free phenolphthalein released from the phenolphthalein-β-D-glucuronidase substrate under the action of intestinal β-GA in 1 h of incubation of the studied material in acetate buffer solution at 37 °C in 1 mL (at pH 4.5–5.6).

The acute toxicity of the glucarolactone–polymer complex was assessed by the method of Prozorovsky [55] on Wistar rats (20 animals) weighing 120–140 g by introducing a freshly prepared solution of the complex in phosphate buffer system intragastrically on an empty stomach. Each substance was tested in 5 doses (4 individuals for each dose): from 1000 to 20,000 mg/kg of animal weight. The number of deaths in the groups was recorded 48 h after the administration of the substances.

In vivo experiments were performed in (National Medicine Research Center of Oncology, St.-Petersburg, Russia) and involved 300 (one hundred for each group, including control) white male and female outbred Wistar rats (NMRC of Oncology, St Petersburg, Russia according to the procedure described in the work [55]). The animal masses varied from 120 to 140 g. Butyl-butanol-nitrosamine (BBNA) (Scientific Research Institute of Organic Intermediates and Dyes, Moscow (NIOPIK)) was used as a carcinogenic agent (2500 mg/kg) that selectively induced bladder tumors. The first (reference) group included the animals to whom only butyl-butanol-nitrosamine (causing bladder tumors in 100% of cases) was administered; both carcinogen and polymer complexes of glucarolactones were administered to the second (experimental) group. The tested preparation (1,4–1,6 lactones bound to the polymer carrier) was given to rats in the form of aqueous solution 3 times a week (dose: 1250 mg/kg).

The study was approved by the local ethical committee (Committee on the Ethics of Animal Experiments of the National Medicine Research Center of Oncology, St.-Petersburg, Russia) and all experiments were performed in accordance with the appropriate recommendations.

## 3. Results and Discussion

### 3.1. Synthesis of Water-Soluble Copolymers of N-Vinylamines

In the majority of cases, the formation of polymer complexes with drugs makes it possible to decrease drug toxicity and to prolong drug action [5,6,7,8]. In this work, copolymers of N-VP with N-VA were selected as polymer carriers for lactones of D(+)-glucaric acid.

The copolymers were obtained by free radical copolymerization of N-vinylpyrrolidone (N-VP) and N-vinylformamide (N-VFA) ([N-VP]_n_–[N-VFA]_m_) followed by the removal of formyl protecting groups and formation of copolymer [N-VP]_n_–[N-VA]_m_ X^−^, where H_2_PO_4_ = X.

The synthesis conditions for precursor copolymers [N-VP]_n_–[N-VFA]_m_ (333 K) and target copolymers [N-VP]_n_–[N-VA]_m_ X^−^ in aqueous solutions are given in Table 1.

The processes of acidic hydrolysis by strong acids (sulfuric, hydrochloric, nitric acid) are well understood [46,56].

The initiator was used in different weight percentages to obtain statistical data on the conversion of the copolymerization process and the resulting molecular weight, which was estimated from the intrinsic viscosity.

Since it is practical to realize complex formation between copolymers and lactones of D(+)-glucaric acid in the phosphate buffer, at the stage of acidic hydrolysis we used an excess of orthophosphoric acid, which does not need to be removed from the system, and the addition of a calculated amount of sodium alkali allows for obtaining a phosphate buffer with pH = 4.5. (Figure 1):

Acidic hydrolysis and the appearance of [N-VP]_n_-[N-VA]_m_ X^−^ were monitored by TLC (detection of the released formic acid). The proposed technique offered the possibility of reliable control of the chromatographic zone of formic acid (retention factor R_f_ = 0.9), since the copolymer [N-VP]_n_-[N-VA]_m_ becomes adsorbed at the start (*R*_f_ = 0).

Molecular mass distributions (MMD), polydispersity, compositional homogeneity, the absence of monomers, and other low molecular weight compounds are important for polymers intended for biomedical applications. The presence of residual monomers in the products of N-VP/N-VA copolymerization was detected by TLC. The TLC chromatographic system was selected on the basis of the theory of eluotropic series (Snyder’s solvent selectivity triangle) [57] and solvent strength (S_i_) for direct-phase TLC (“Prizma” model) [58]. The ternary eluent (chloroform (S_i_ = 4.1), ethyl acetate (S_i_ = 4.4), and aqueous solution of ammonia (S_i_ = 10.2)) was selected in such a way that the mobility of the [N-VP]_n_–[N-VFA]_m_ copolymer under these conditions was equal to zero. Thus, it was possible to reliably register the presence of residual monomers N-VP (R_f_ = 0.84) and N-VFA (R_f_ = 0.58) in the reaction medium.

It is known that some restrictions (caused by the necessity of biodegradation) are placed on molecular masses of carbochain polymeric carriers of BAC. Molecular masses of medicinal polymers included in preparations for internal use should not exceed 40 × 10^−3^ g∙mol^−1^ [59,60]. However, complexes with GL are intended for the prevention of stomach and bladder cancer; it is suggested to administer them perorally. Thus, the restrictions concerning molecular masses of polymers are inessential for (N-VP-N-VA) carriers.

### 3.2. Determination of Molecular and Structural Characteristics of Copolymers

Molecular mass distributions (MMD), polydispersities ($\bar{M}$*_w_*$/\bar{M}$*_n_*), and average molecular masses (AMM) were determined by exclusion (gel-permeating) chromatography.

Today, high-performance liquid chromatography (HPLC) is among the most important methods for the analysis of molecular characteristics of synthetic and natural polymers [61,62]. Theoretical basics and principles of practical application of this method for flexible chain polymers are well known [63,64,65,66,67].

Since the Mark–Kuhn–Houwink constants in the equation [*η*] = K∙M^α^ for random copolymers of this class have not been determined [68,69], “relative” AMM and MMD were calculated using the known Mark–Kuhn–Houwink constants in 0.2 M NaCl solution for one block of a copolymer (when it predominates in a copolymer, 75 mol.% or less). The following values were used in this work: [*η*]_PVFA_ = 10.74 × 10^−3^ M^0.76^ [40]; [*η*]_PVP_ = 14.1 × 10^−3^ M^0.70^ [70].

The chromatographic system was calibrated using the PVFA standards obtained and characterized in our earlier works in 0.2 M NaCl solution in water. When N-vinylpyrrolidone units prevailed in the copolymer, the calibration dependence for PVP (which is necessary to calculate the “relative” MM of a copolymer) was plotted using the universal Benoit calibration dependence [68,69].

Hydrodynamic methods (velocity sedimentation, diffusion, dynamic light scattering, DLS) were used as independent methods to confirm the reliability of the above technique for AMM determination. For a series of copolymers, weight-average molecular masses ($\bar{M}$*_w_ **) were determined by dynamic static light scattering (DLC) (Table 2). The comparison of weight-average MM obtained by HPLC, velocity sedimentation, and light scattering $(\bar{M}$*_w_ **) shows that these values agree reasonably well.

In the case of the copolymer containing 50 mol.% of N-VA, the discrepancy between the $\bar{M}$*_w_*
*** values determined by HPLC and DLS are more appreciable. For the determination of $\bar{M}$*_w_*, significant values of 3–5 points were measured both for SEC and DLS methods. The possible cause of this variance is an uncertainty of calculations of “relative” MM and, in addition, a higher possibility of formation of supramolecular structures. Large associates are formed due to the decreased strength of non-specific interactions between carbonyl groups and amino groups in dilute salt solutions, when the percentage of amino groups in a macromolecule increases. However, even if the N-VA content in the copolymer is equal to 50 mol.%, the relative number of supramolecular structures in the solution (according to the DLS data) is lower than 1%. Apparently, the existing associates cannot be registered by HPLC and sedimentation analysis due to low amounts of NH_2_ groups. However, it can be stated that for random water-soluble copolymers of [N-VP]_n_-[N-VA]_m_ (when the percentage of the second component does not exceed 15 mol.%) this calculation of the “relative” AMM may be useful. On the whole, the MM and polydispersities of the obtained series of copolymers are relatively small and meet the requirements imposed on polymeric carriers for medicinal preparations [9,12,16,22].

### 3.3. Preparation of Polymer-Based Complexes with Glucarolactones

The presence of reactive protonated amino groups in the copolymers facilitates specific interactions between these copolymers and glucarolactone, which lead to the formation of polymer-based complexes. These complexes were prepared by stirring the solution of copolymer [N-VP]_n_-[N-VA]_m_ in phosphate buffer together with solutions of individual 1,4- and 6,3-lactones for 40 h at room temperature. The excess glucarolactones were separated by dialysis, and the product was lyophilized.

Quantitative TLC analysis established that the portion of bound lactones increased with increasing duration of interaction between lactones and the polycation (Figure 2).

The ability of lactones to inhibit β-GA depends on their stereoisomeric composition. It has been established [70] that the activities of lactones decrease in the following order: 1,4-glucarolactone > 1,4-6,3-diglucarolactone > glucaric acid > 6,3-glucarolactone > glucuronic acids > glucano-D-lactone > glucuronolactone. The most efficient inhibitor of β-GA is the individual 1,4-GL; however, it turned out that this inhibiting effect lasts only for several hours.

It has been established that upon peroral administration to rats, the mixture of 1,4- and 6,3-GL had only an insignificant effect on the inhibition of β-glucuronidase (β-GA) in urine (Figure 3), which agrees with the earlier results [71]. The activity of individual lactones and the [N-VP]_n_-[N-VA]_m_-[GL] complex was determined according to the Fishman procedure [54] by the amount of released phenolphtaleine-glucaronide (substrate) under the action of β-GA. However, an attempt was made to isolate individual glucarolactones and to prepare solutions enriched with 1,4 and 6,3-GL, which were further used in the synthesis of the polymer-based complex.

When isolating individual 1,4- and 6,3-GL, it is necessary to take into account the following facts: in an aqueous solution of D(+)-glucaric acid, the equilibrium exists between several compounds (acyclic compound (1) - 1,4-glucarolactone, (2) - 6,3-glucarolactone (3), and 1,4-6,3-glucarodilactone (4), Figure 4). During the lactonization of glucaric acid, 6,3-GL appears first, and 1,4-GL is formed slower. Upon reaching equilibrium, the mixture contains 40% of glucaric acid, 30% of 1,4-GL, and 30% of 6,3-GL 1,4-6,3-glucarodilactone was not found under these conditions and appeared only at higher temperatures [72].

It has been demonstrated that in an aqueous solution (pH = 7), the content of 1,4-GL gradually decreased down to 80% in 5.5 h (with respect to its initial concentration); equal amounts of glucaric acid and 6,3-GL were formed [73]. The addition of a strong acid promotes reaching equilibrium and changes the ratio between amounts of the resulting products in favor of the formation of D(+)-glucaric acid.

The attempt to obtain individual 1,4-GL and 6,3-GL [48] demonstrated that the content of 1,4-GL in the dry mixture decreased down to 40% as a result of isomerization of 1,4-GL into 6,3-GL at elevated temperatures. The mixture of lactones enriched with 1,4-GL was obtained by extraction with aliphatic alcohols followed by the distilling of alcohols under a vacuum at 298 K; 6,3-GL was isolated by crystallization from the aqueous layer (Table 3). Individual 1,4-GL was isolated from the concentrated alcohol solution with a density of 1.36 g/cm^3^. The isolated target 1,4-GL and 6,3-GL had the melting points of 366 K and 422 K, respectively (which corresponded to the literature data).

#### Investigations of Formation of Polymer/Glucarolactone Complexes by Spectral and Chromatographic Methods

Chromatographic (TLC, HPLC) methods were developed for rapid determination of the content of lactones of D(+)-glucaric acid in the process of preparation of their polymer complexes.

The TLC results indicated that in weakly alkaline eluent (pH 7.8–7.9), the polymer/glucarolactone complex is relatively strong (Figure 5).

The eluents for TLC were selected after analysis of chromatographic activity of solvents (Snyder’s solvent selectivity triangle). The selected chromatographic system *n*-butanol/acetic acid/water (2:1:1) allows one to separate, identify, and quantitatively determine the percentages of components of the final mixture of products. It is seen from the TLC data that under these conditions, the formed polymer-based complexes with 1,4- and 6,3-lactones can be separated into a number of components.

The chromatographic zone with *R*_f_ = 0.54 is attributed to 6,3-glucarolactone; the zone with *R*_f_ = 0.78 is related to 1,4-lactone, which was confirmed by the TLC experiment involving pure 1,4- and 6,3-lactones as reference compounds. Under these conditions, the polymer complex releases free GL, and the carrier becomes adsorbed at the start (*R*_f_ = 0, Figure 6). Decomposition of the complex at pH = 5.5 and release of free GL are extremely important for in vivo inhibition of β-GA, whose activity is especially high in weakly acidic media. The TLC chromatograms presented in Figure 6 show that, upon prolonged storage, two new components appear in the polymer complex (*R*_f_ = 0.6 and *R*_f_ = 0.44), possibly 1,4-6,3-dilactone and D(+)-glucaric acid.

Eluent*:*
*n*-butanol/acetic acid/water (2:1:1), pH = 5.5. Detection: iodine vapors. Sensitivity of the method: 0.25 µg.

The formation and structures of polymer-based GL complexes were confirmed by spectral methods. The ^1^H NMR spectrum of the complex (Figure 7) contains the broadened peaks attributed to copolymers [N-VP]_n_-[N-VA]_m_ (d) and the signals that may be assigned to the equilibrium mixture of glucarolactones and glucaric acid. In area (a) (5 ppm), separate signals are observed that correspond to the proton of the methine group localized near the oxygen atom of the 1,4-glucarolactone ring. The signals in area (c) 3.9–4.4 ppm are related to glucaric acid protons. The area (b) 4.5–4.7 ppm contains complex multiplets that can be attributed to interactions between the amino group of the copolymer and the carboxyl group of glucarolactone and sodium phosphates.

To confirm the existence of interaction between 1,4-glucarolactone and the amino group of the copolymer, their stoichiometric mixtures were prepared (with and without sodium phosphates). In the ^1^H NMR spectrum of the mixture without the addition of phosphates, the signal attributed to the methine group near the carboxylic group shifts upfield by 0.3 ppm, which indicates interaction between the carboxylic group and the amino group of the copolymer (Figure 8).

The downfield shift of the signals related to protons of 1,4-GL adjacent to carbonyl atoms after substituting K^+^ for H^+^ is caused by diminished shielding [72].

The complex multiplet (b) is located on the shoulder of the residual water peak corresponds to protons of 1,6-GL and it is impossible to evaluate its content. However, using of size exclusion chromatography allows to determine it exactly A typical chromatographic profile of the polymer-based complex is presented in Figure 9.

The SEC profile (Figure 8) clearly demonstrates the existence of dynamic equilibrium between [N-VP]_n_-[N-VA]_m_ [1,4- and 6,3-GL] complex and different forms of glucarolactones (1,4-GL, 6,3-GL) and D–glucaric acid. The assignment of peaks in the chromatogram was carried out using reference compounds. Table 4. presents the data on the relationships between the amounts of glucaric acid forms in the complex.

Thus, the ^1^H NMR and size exclusion chromatography data demonstrate that the presence of phosphates makes it possible to decrease the content of glucaric acid in medicinal substances.

IR spectroscopy was used to confirm the interaction between 1,4-glucarolactone, the copolymer, and sodium phosphates. The IR spectrum of the mixture of anhydrous phosphates involves the following four absorption bands: 1150 (ν_s_P = O), 1057 (ν_as_P-O), 933 (ν_s_P-O), and 860 (νP-O-H) cm^−1^ [74]. The following changes were observed in the IR spectrum of the mixture of copolymer [N-VP]_n_-[N-VA]_m_ with sodium phosphates: an increase in the intensity of the bands corresponding to P–O valence vibrations, an insignificant shift, and a narrowing of peaks. These features are indicative of interaction between sodium phosphates and carbonyl groups of lactam rings. The C=O vibration band of the lactam ring shifts slightly (by 15 cm^–1^) to the low frequency region (see Appendix A). Several spectral changes were also revealed by analysis of the spectrum of the initial 1,4-GL and the spectrum of the complex. The absorption bands in the area of the valence vibrations of carbonyl and carboxyl groups of 1,4-glucarolactone (1771 and 1722 cm^−1^, respectively) broaden and become slightly shifted toward the high-frequency region. An insignificant shift (by 22 cm^–1^) of the C=O absorption band toward lower frequencies (Appendix A) confirms the interaction between 1,4-glucarolactone and the lactam ring of the copolymer.

Based on the analysis of spectral and chromatographic data, it can be concluded that the glucarolactone/polymer complex contains the mixture of 1,4- and 6,3-GL of glucaric acid. Apparently, it is formed at the expense of ionic interactions between charged groups of the polymer (amino group) and lactone (carboxylic group); besides, the phosphoric acid residue and multiple hydrogen bonds between the lactam ring of the VP unit of the copolymer and the hydroxyl groups of lactams also take part in this interaction.

The research performed in this work made it possible to optimize the preparation method for a biologically active polymer-based product (Figure 9). This product is a complex between the N-vinylpyrrolidone/N-vinylamine copolymer and the equimolar mixture of 1,4- and 6,3-lactones of D(+)-glucaric acid (1 mol) stabilized by sodium phosphates Na_2_HPO_4_ (0.1 mol) and NaH_2_PO_4_ (0.1 mol); the ratio between units is 4:1 (0.46 mol).

### 3.4. Investigation of Inhibiting Action and Stability of Complex

The inhibiting action of the phosphate complex with [N-VP]_n_-[N-VA]_m_-[1,4-6,3-GL] toward enzyme β-glucuronidase was studied in vitro. The concentrations necessary for 50% suppression of the enzyme activity (the end point infective doses ID_50_, mg/mL) were established for complex and individual 1,4-GL of D(+)-glucaric acid (Table 5).

Substrate: phenolphthalein-D-glucuronide, enzyme concentration [E] = 3 Fishman units (3 ng); T = 310 K; incubation time: 1 h; pH 10.2–10.3; λ = 540 nm.

Stability of the complexes and their ability to inhibit activity of β-GA weakly depended on the MM of the copolymer within the 30 ÷ 80 kDa range. However, further increase in the MM resulted in a decrease in biological activity (see Table 6).

Of special interest is the in vivo investigation of the influence of dynamics on the development of bladder tumors upon introducing the carcinogen butyl-butanol-nitrosamine (BBNA) to rats (Table 7). The experiments involved two groups of rats. The first group of animals received only the carcinogen, and the other group was given both the carcinogen and preparation. In the first group of animals, the enzyme activity grew by almost four times (from 62 to 244 Fishman units). In the second group of animals, the enzyme activity was maintained at a constant (2.8 Fishman units).

The averaged data on the activity of β-GA in the 24-h rat urine at different times after the beginning of the experiment are shown in Table 7.

All the animals of the first group (to whom only the carcinogen was administered) died in 10–12 months; bladder carcinomas were found in all rats. On the contrary, about 80% of the rats of the second group (who took both the carcinogen and complex) survived by the 12th month of the experiment; the first precancerous changes were observed only during the 17th month of the experiment.

As seen from the data presented in Table 7, the introduced BBNA caused a dramatic increase in the enzyme activity (up to 348 units by the 11th month of the experiment, a threefold increase from the 3rd to the 11th months). An increase in the β-glucuronidase content was accompanied by accelerated development of papillomas and bladder tumors. The activity of the enzyme in the rats of the second group (the carcinogen + complex) during the whole experiment was strongly suppressed and varied from 3 to 6 units (normal activity of β-glucuronidase ranges from 100 to 120 Fishman units).

The acute toxicity of the complex was studied using outbred rats. The maximal tolerated dose (MTD) was established after peroral administration to rats and turned out to be 20,000 mg/kg.

The LD_50_ value for [N-VP]_n_-[N-VA]_m_-[1,4-6,3-GL] could not be determined, since the lethality in the group of animals with a maximum dose of 20,000 mg/kg was below 50%. This made it possible to classify the glucarolactone-polymer complex as a relatively harmless substance according to the classification of Hodge and Sterner [75] or a class 5 according to the Globally Harmonized System of Classification and Labelling of Chemicals [76].

## 4. Conclusions

Water-soluble complexes of 1,4- and 6,3-lactones of D(+)-glucaric acid with a N-vinylpyrrolidone/N-vinylamine copolymer were synthesized. The values of the average molecular masses (AMM) of random water-soluble N-vinylamide copolymers determined by different methods (high-performance liquid chromatography, hydrodynamic methods) were compared. It was established that the AMM of random copolymers [N-VP]_n_–[N-VA]_m_ containing no more than 15 mol.% of N-VA units can be calculated from the Mark–Kuhn–Houwink constants for poly-N-vinylpyrrolidone.

Stable complexes of the [N-VP]_n_–[N-VA]_m_ copolymer with lactones of D(+)-glucaric acid were prepared; the products efficiently inhibited the enzyme β-glucuronidase and demonstrated prolonged action after intragastric administration to animals.

Thus, the present work has laid the foundation for the development of polymer-based prophylactic antitumor preparation, which is a specific prolonged-action inhibitor of β-glucuronidase present in urine. The preparation precludes decomposition of the carcinogen/glucuronide conjugates by β-glucuronidase, provides renal excretion of carcinogenic metabolites in the inactive bound form, and prevents the appearance and growth of bladder tumors.

## Figures and Tables

**Figure 1 polymers-14-00105-f001:**
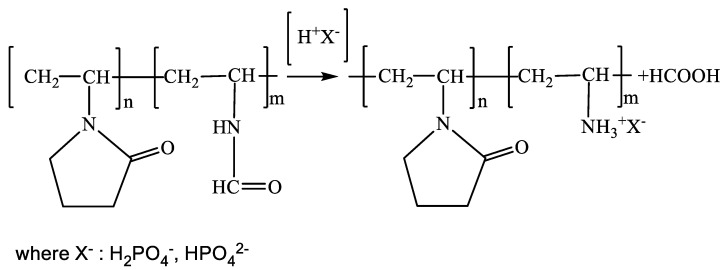
Scheme of acidic hydrolysis of copolymer [N-VP]_n_-[N-VFA]_m_ by phosphoric acid.

**Figure 2 polymers-14-00105-f002:**
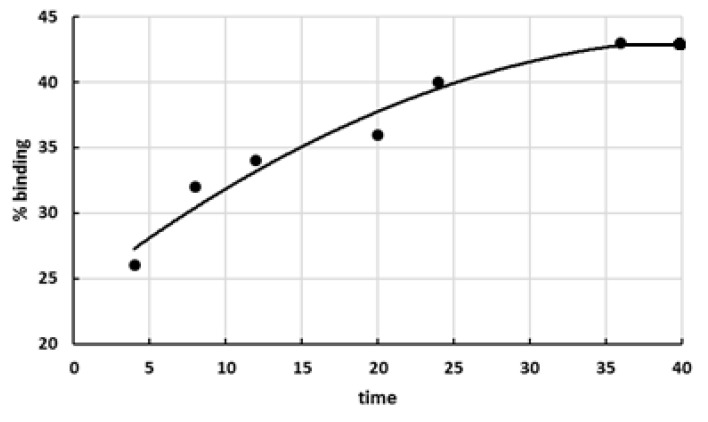
Dependence of the portion of bound lactones of D(+)-glucaric acid on duration of interaction between the polymer carrier and glucarolactones (295 K, phosphate buffer pH 4.75).

**Figure 3 polymers-14-00105-f003:**
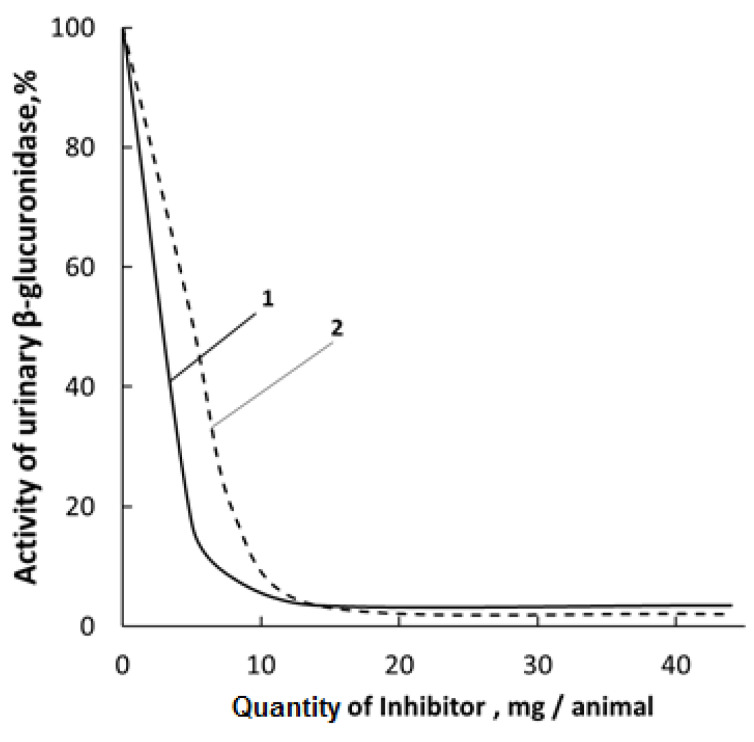
Comparison of activities of 1,4-glucarolactone and the polymer complex against β-glucuronidase present in rat urine: 1-1,4-GL; 2–racemic mixture of 1,4- and 6,3-GL.

**Figure 4 polymers-14-00105-f004:**
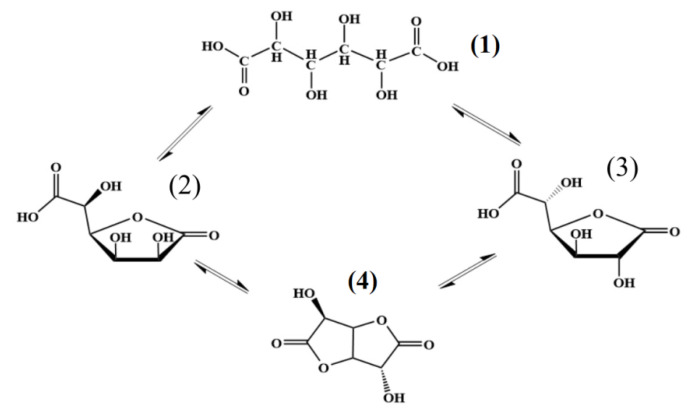
Equilibrium existing in aqueous solution of D(+)-glucaric acid.

**Figure 5 polymers-14-00105-f005:**
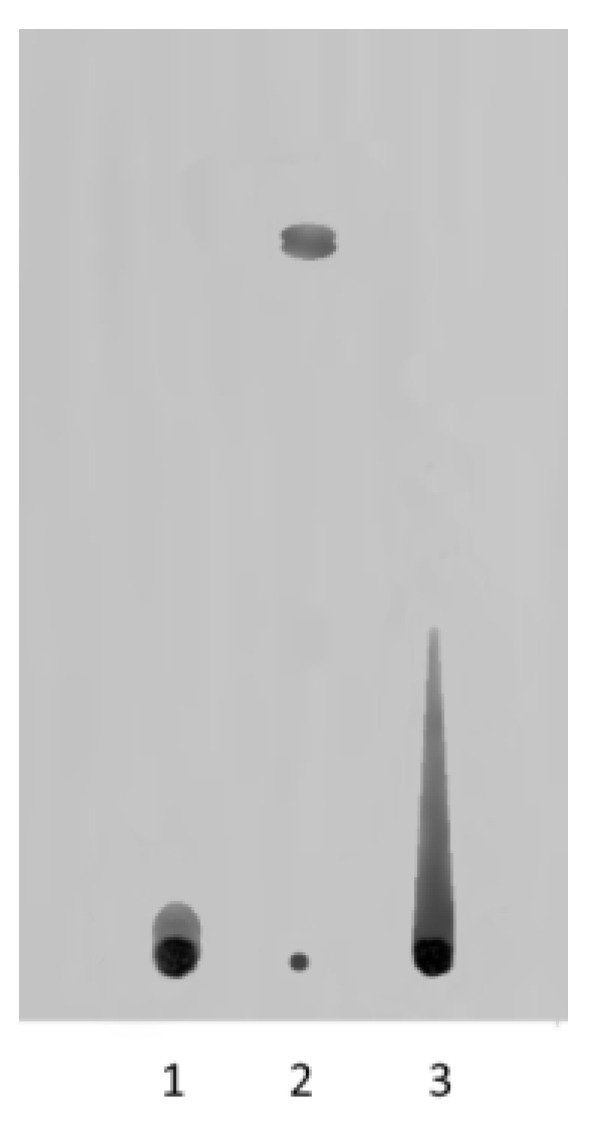
TLC chromatogram of complex [N-VP]_n_-[N-VA]_m_ [1,4- and 6,3-GL]: 1) copolymer [N-VP]_n_-[N-VA]_m_; 2) mixture of 1,4- and 6,3-GL; 3) polymer complex [N-VP]_n_ -[N-VA]_m_ [1,4- and 6,3-GL]. Eluent: ethanol 25%/aqueous solution of ammonia/water (8.0:0.4:1.6). Detection: UV, λ = 365 nm.

**Figure 6 polymers-14-00105-f006:**
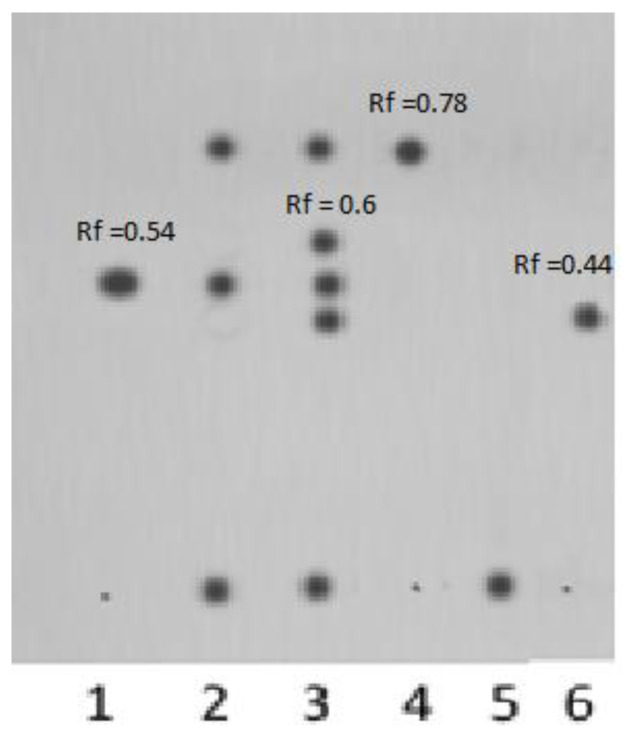
TLC chromatogram of polymer complex [N-VP]_n_-[N-VA]_m_ [1,4- and 6,3-GL]: (1) 6,3-GL; (2) polymer complex after dialysis; (3) sample 2 after prolonged storage at *t* = 23 °C; (4) 1,4-GL; (5) polymer carrier [N-VP]_n_-[N-VA]_m_; (6) D(+)-glucaric acid.

**Figure 7 polymers-14-00105-f007:**
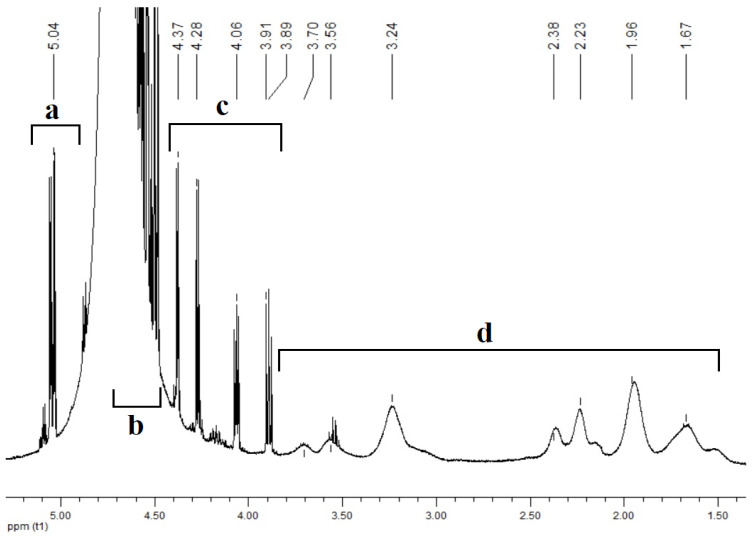
^1^H NMR spectrum of the complex [N-VP]_n_-[N-VA]_m_ [1,4- and 6,3-GL]. Area (a) (5 ppm), (b) 4.5–4.7 ppm, (c) 3.9–4.4 ppm, (d) 1.5–3.8 ppm.

**Figure 8 polymers-14-00105-f008:**
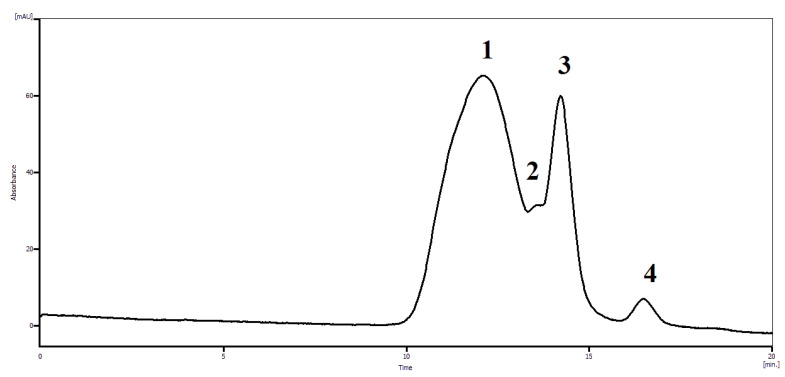
Characteristic chromatographic profile of dynamic equilibrium of [N-VP]_n_-[N-VA]_m_ [1,4- and 6,3-GL] complex: (1) [N-VP]_n_-[N-VA]_m_ [1,4- and 6,3-GL] complex, (2) D–glucaric acid, (3) 1,4-GL, (4)–6,3-GL. Column: waters Ultrahydrogel linear, flow rate: 0.7 mL/min, eluent: 0.2 M NaCl, 237 K.

**Figure 9 polymers-14-00105-f009:**
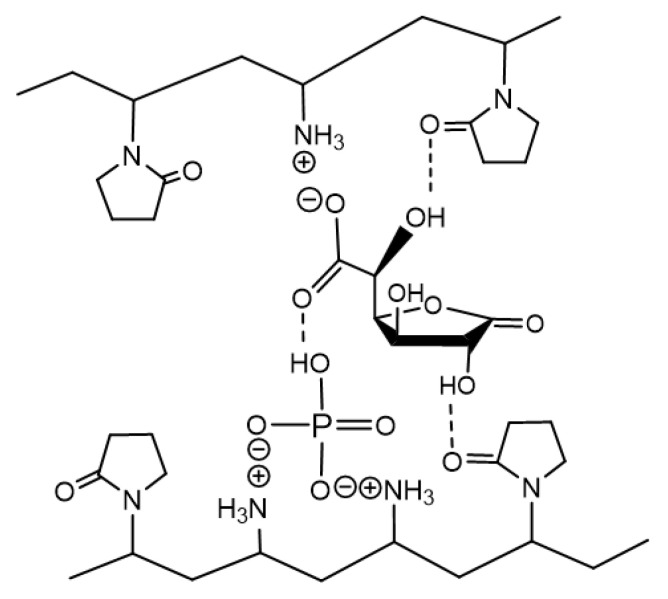
Proposed structure of the complex between the copolymer [N-VP]_n_-[N-VA]_m_ and [1,4- and 6,3-GL] lactones.

**Table 1 polymers-14-00105-t001:** Conditions of radical copolymerization of N-VP with N-VFA in the presence of NH_4_OH and H_2_O_2_; preparation of copolymer [N-VP]_n_–[N-VA]_m_ X^–^ by acidic hydrolysis with orthophosphoric acid.

№	Copolymerization of N-VP and N-VFA						Removal of Formyl Protection			
	N-VP/N-VFA Ratio, mol.%	Monomer Content, wt.%	H_2_O_2_, wt.% with Respect to the Total Amount of Monomers	NH_4_OH wt.% with Respect to the Total Amount of Monomers	t, h	[η]; 238 K, 0.1M KCl, cm^3^/g	Copolymer Content in Solution, wt.%	H_3_PO_4_ Content wt.%	T, K	t, h
1	90:10	46.6	0.8	2.0	4	36	30	13.70	383	24
2	89:11	50	0.5	1.37	4	24	10	7.33	383	24
3	91:9	35	0.6	1.8	5	15	31	7.78	373	24
4	50:50	30	0.8	1.4	4	29	10	5.50	373	20

**Table 2 polymers-14-00105-t002:** Hydrodynamic and molecular characteristics of copolymers [N-VP]_n_-[N-VA]_m_ in 0.2 M NaCl.

№	N-VP, mol.%	[*η*], cm^3^/g	*D* × 10^−7^, cm^3^/g	*R*_h_, nm	*M*_SD_, kDa	$\bar{M}$_*w*_ *, kDa	$\bar{M}$_*w*_ kDa	${\bar{M}}_{w}/{\bar{M}}_{n}$
1	90	32.0	4.6	5.7	50.0	54.0	60.0	2.3
2	89	25.0	5.2	4.1	31.0	36.0	40.0	2.2
3	91	19.0	5.9	3.8	27.0	28.0	25.0	2.1
4	50	28.0	5.3	4.2	48.0	42.0	50.0	2.2

*R*_h_ = hydrodynamic radii (determined from the DLS data); [*η*] = intrinsic viscosity (determined from the viscosimetry data); *M*_SD_ = molecular mass determined from the velocity sedimentation data; *M*_n_ = number-average molecular mass (determined from the DLS data); $\bar{M}$*_w_* = weight-average molecular mass (determined from the DLS data); $\bar{M}$*_w_*
*** = weight-average molecular mass (determined from the SEC data); *M*_w_/*M*_n_ is the polydispersity index.

**Table 3 polymers-14-00105-t003:** Extraction of individual glucarolactones.

№	Extracting Agent *	Content of 1,4-GL in Alcohol Solution, wt.%	Content of 6,3-GL in Aqueous Phase, wt.%
1	Butanol-1	6.5	84.3
2	Tert-butanol	5.9	67.5
3	Tert-amyl alcohol	9.3	31.3

* Temperature of alcohol distillation: 308 K.

**Table 4 polymers-14-00105-t004:** Contents of different forms of glucaric acid in [N-VP]_n_-[N-VA]_m_ [1,4- and 6,3-GL] complex.

№	Substance	Retention Time, min	Content of a Substance in Complex, %	Content of a Substance in Complex (without Phosphates), %
1	D(+)-glucaric acid	13.70	17.7	25.7
2	1.4-glucarolactone	14.37	78.1	72.8
3	6.3-glucarolactone	16.42	4.2	1.5

**Table 5 polymers-14-00105-t005:** ID_50_ of complex and 1,4-GL.

Substance	ID_50_, mg/mL
1,4-lactone of glucaric acid	0.005
Complex	0.06

**Table 6 polymers-14-00105-t006:** Influence of copolymer composition and its molecular mass on specific activity and stability of complexes.

№	Characteristics of the Copolymer (Polyelectrolyte)		Binding of Lactones (%)	Stability of Complex in Aqueous Solution (24 h), %	Residual Activity of Enzyme β-GA, Fishman Units
	Content of VA Units, mol.%	MM × 10^3^,Da			
1	15	80	90	85	<10
2	20	30	84	90	<10
3	20	60	90	88	<10
4	20	100	90	91	<10
5	20	150	95	80	20…40

**Table 7 polymers-14-00105-t007:** Activity of β-GA in 24-h rat urine depending on duration of the experiment.

Groups of Animals	Introduced Compounds		Activity of β-GA (Fishman Units) in:						
		1 Day	3 Months	6 Months	9 Months	12 Month	18 Months	24 Months	28 Months
1	BBNA	0.3	112 ± 1.6	120 ± 2.4	268 ± 4.2	348 + 6.7 *			
2	BBNA + complex	0.3	4.0 ± 0.3	6.0 + 0.4	5.0 ± 0.4	3.0 + 0.2	4.2 + 0.3	5.4 + 0.7	6.2 + 0.8

* 11 months.

## Data Availability

The data presented in this study are available on request from the corresponding author.

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
