# Peer review of "Complexes of Glucarolactones with Water-Soluble Copolymers of N-Vinylpyrrolidone with N-Vinylamine as Inhibitors of β-Glucuronidase Efficacy"

_polymers, 2021, doi:10.3390/polym14010105_

Round 1

Reviewer 1 Report

Herein, the authors conducted a synthesis of N-vinylpyrrolidone copolymers and characterized them spectroscopically and for their ability to inhibit the enzyme b-glucuronidase.

The study somewhat basic but the authors have employed an incredible amount of detail.

The title should include something along the lines of: 'and their b-glucuronidase inhibition efficacy'

The authors should also make use of an ESI file -- much of the materials and methods along with some of the latter figures (NMR and FTIR) should be contained within.

Author Response

Dear Reviewer,

The study somewhat basic but the authors have employed an incredible amount of detail.

1.The title should include something along the lines of: 'and their b-glucuronidase inhibition efficacy'

The title had been changed

2. The authors should also make use of an ESI file -- much of the materials and methods along with some of the latter figures (NMR and FTIR) should be contained within.

FTIR data was attached in Supporting Information file.

Reviewer 2 Report

Comments on the research article title: Complexes of glucarolactones with water-soluble copolymers of N-vinylpyrrolidone

Manuscript ID: polymers-1516876 

The authors present a research paper on Water-soluble complexes of N-vinylpyrrolidone/N-vinylamine copolymers with lactones of D-glucuronic acid. The complexes were prepared and characterized though various methods and demonstrated enzyme inhibition activity in vitro and in vivo. The research was well designed; however, there are some typo, grammar, and punctuation errors which authors could track and make changes. Also, questions and comments I would like to make are as follows.

  1. The title can be and be adjusted to be more specific since only copolymer of N-vinylpyrrolidone/N-vinylamine copolymers were made and not with other monomers.
  2. The keywords were too general and can be made more specific (HPLC, NMR, light scattering, electronic and vibrational spectroscopy, diffusion analysis should be removed).
  3. Subtitles should be made for heading 2. Materials and methods, so that readers can follow your experiments and understand the objective of each test.
  4. Line 98-110, more details of the synthesis process and structural characterization (NMR, FTIR, TLC) should be described so that the experiment can be reproduced.
  5. Line 100, [N-VFA] should be mentioned prior to presenting the abbreviation.
  6. Line 103-104, why was the ratio of acid and N-VFA of 4:1 used.
  7. According to the in vivo experiment, was there any ethical approval prior to performing the experiment. This should be mentioned in the method section. Also, more details on the experiment are required e.g., number of rats involved in each group, how was the sample prepared (concentration, solvent used), was the control group given anything in place of the sample?
  8. Since there is no biocompatibility test performed, what are your suggestions on the toxicity of the synthetic preparation made?
  9. Table 1, Please describe your idea on varying the monomers mol ratio. Also, why was the initiator (H2O2) used at different wt% for each experiment? Would this affect the synthesis and be comparable?
  10. Please consider re-writing Line 281-284, the information provided is quite confusing of which data were from HPLC and which from the other.
  11. Line 289, was there any statistical analysis made to state the significance. (How many repetitions were performed for each experiment? This should also be presented.)
  12. Line 306-308, please indicate the amount or concentration of polymer and lactones used.
  13. Figure 2, since the binding was performed up to 40 h, why was the binding at 40 h not plotted.
  14. The method for the experiment done for Figure 3 should be added to the method section. The x-axis of Figure 3 should be revised. Shall it be the “amount” of inhibitor, not “concentration”?
  15. Line 324, researcher’s name, lab group and year performed should be mentioned and not the nationality.
  16. Figure 6, Rf labeling should be added for a better understanding of the image. Line 386-387, should be placed with the figure caption.
  17. More description should be added for Table 5. (Line 419-420)
  18. Line 475-479, the information provided does not seem to match the result shown in Table 8 (4 times, 62-244 units?). Is there any information on the baseline (0 months) of B-GA for both groups?
  19. Please add the reference for the information on Line 301 and Line 496.

Author Response

Dear Revier

  1. The title can be and be adjusted to be more specific since only copolymer of N-vinylpyrrolidone/N-vinylamine copolymers were made and not with other monomers.  

The title had been changed.

  1. The keywords were too general and can be made more specific (HPLC, NMR, light scattering, electronic and vibrational spectroscopy, diffusion analysis should be removed).

The keywords had been changed

2. Subtitles should be made for heading 2. Materials and methods, so that readers can follow your experiments and understand the objective of each test.

Experimental section was divided into subsections

3. Line 98-110, more details of the synthesis process and structural characterization (NMR, FTIR, TLC) should be described so that the experiment can be reproduced.

Spectral data for synthesized copolymers was added (lines 184-18)

4. Line 100, [N-VFA] should be mentioned prior to presenting the abbreviation.

Full name was added (lines 161, 163)

5. Line 103-104, why was the ratio of acid and N-VFA of 4:1 used.

explanation was given at lines 275-279

6. According to the in vivo experiment, was there any ethical approval prior to performing the experiment. This should be mentioned in the method section. Also, more details on the experiment are required e.g., number of rats involved in each group, how was the sample prepared (concentration, solvent used), was the control group given anything in place of the sample?

Ethical regulations and amount of introduced caccerogen was added  (lines 246-249)

7. Since there is no biocompatibility test performed, what are your suggestions on the toxicity of the synthetic preparation made?

Toxicity tests were performed, but not included in previous version of the manuscript. Preparation appeared to be non toxic (lines 230-234)

8. Table 1, Please describe your idea on varying the monomers mol ratio. Also, why was the initiator (H2O2) used at different wt% for each experiment? Would this affect the synthesis and be comparable?

Explanation were added (lines 272-274)

9. Please consider re-writing Line 281-284, the information provided is quite confusing of which data were from HPLC and which from the other.

Explanations were added (lines 332-339) and below.

10. Line 289, was there any statistical analysis made to state the significance. (How many repetitions were performed for each experiment? This should also be presented.) 

3-5 repetitions were done and the average values are presented in the table (line 342)

11. Line 306-308, please indicate the amount or concentration of polymer and lactones used.

Corrections were itrodused in experimental section (lines 215-220)

12. Figure 2, since the binding was performed up to 40 h, why was the binding at 40 h not plotted.

Corrections were made.

13. The method for the experiment done for Figure 3 should be added to the method section. The x-axis of Figure 3 should be revised. Shall it be the “amount” of inhibitor, not “concentration”?

Corrections were made.

14. Line 324, researcher’s name, lab group and year performed should be mentioned and not the nationality.

It was corrected.

16. Figure 6, Rf labeling should be added for a better understanding of the image. Line 386-387, should be placed with the figure caption.

Rf were added.

17. More description should be added for Table 5. (Line 419-420)

Description were added ( lines 475-478).

18. Line 475-479, the information provided does not seem to match the result shown in Table 8 (4 times, 62-244 units?). Is there any information on the baseline (0 months) of B-GA for both groups?

Zero point (base line) (at 1 day) was added to the table 7.

19. Please add the reference for the information on Line 301 and Line 496.

Corresponding references were added.

Round 2

Reviewer 2 Report

The manuscript was substantially improved. However, there are minor changes that I think the authors could make prior to the publication.

  1. The period (.) should be removed from the title.
  2. Line 474, Fig.8 should be Figure 8 to maintain consistency throughout the manuscript.
  3. Line 507 and 508 mentioned Figure 14 and Figure 12 which I believe do not exist in the manuscript. The appropriate figure reference should be made.

Author Response

he manuscript was substantially improved. However, there are minor changes that I think the authors could make prior to the publication.

  1. The period (.) should be removed from the title.

Corrected

2. Line 474, Fig.8 should be Figure 8 to maintain consistency throughout the manuscript.

Corrected

3. Line 507 and 508 mentioned Figure 14 and Figure 12 which I believe do not exist in the manuscript. The appropriate figure reference should be made.

Yes, it was a mistake, which is corrected
